# Generalized Natural Language Grounded Navigation via Environment-agnostic Multitask Learning

## Abstract

Recent research efforts enable study for natural language grounded navigation in photo-realistic environments, e.g., following natural language instructions or dialog. However, existing methods tend to overfit training data in seen environments and fail to generalize well in previously unseen environments. In order to close the gap between seen and unseen environments, we aim at learning a generalizable navigation model from two novel perspectives: (1) we introduce a multitask navigation model that can be seamlessly trained on both Vision-Language Navigation (VLN) and Navigation from Dialog History (NDH) tasks, which benefits from richer natural language guidance and effectively transfers knowledge across tasks; (2) we propose to learn environment-agnostic representations for navigation policy that are invariant among environments, thus generalizing better on unseen environments. Extensive experiments show that our environment-agnostic multitask navigation model significantly reduces the performance gap between seen and unseen environments and outperforms the baselines on unseen environments by 16% (relative measure on success rate) on VLN and 120% (goal progress) on NDH, establishing the new state of the art for the NDH task.

## 1 Introduction

Navigation in visual environments by following natural language guidance (Hemachandra et al., 2015) is a fundamental capability of intelligent robots that simulate human behaviors, because humans can easily reason about the language guidance and navigate efficiently by interacting with the visual environments. Recent efforts (Anderson et al., 2018b; Das et al., 2018; Thomason et al., 2019) empower large-scale learning of natural language grounded navigation that is situated in photo-realistic simulation environments.

Nevertheless, the generalization problem commonly exists for these tasks, especially indoor navigation: the agent usually performs poorly on unknown environments that have never been seen during training. One of the main causes for such behavior is data scarcity as it is expensive and time-consuming to extend either visual environments or natural language guidance. The number of scanned houses for indoor navigation is limited due to high expense and privacy concerns. Besides, unlike vision-only navigation tasks (Mirowski et al., 2018; 2016; Xia et al., 2018; Manolis Savva* et al., 2019; Kolve et al., 2017) where episodes can be exhaustively sampled in simulation, natural language grounded navigation is supported by human demonstrated interaction and communication in natural language. It is impractical to fully collect and cover all the samples for individual tasks.

Therefore, it is essential though challenging to efficiently learn a more generalized policy for natural language grounded navigation tasks from existing data (Wu et al., 2018a;b). In this paper, we study how to resolve the generalization and data scarcity issues from two different angles. First, previous methods are trained for one task at the time, so each new task requires training a brand new agent instance that can only solve the one task it was trained on. In this work, we propose a generalized multitask model for natural language grounded navigation tasks such as Vision-Language Navigation (VLN) and Navigation from Dialog History (NDH), aiming at efficiently transferring knowledge across tasks and effectively solving both tasks with one agent simultaneously.

Moreover, although there are thousands of trajectories paired with language guidance, the underlying house scans are restricted. For instance, the popular Matterport3D dataset (Chang et al., 2017) contains only 61 unique house scans in the training set. The current models perform much better in seen environments by taking advantage of the knowledge of specific houses they have acquired over multiple task completions during training, but fail to generalize to houses not seen during training. Hence we propose an environment-agnostic learning method to learn a visual representation that is invariant to specific environments but still able to support navigation. Endowed with the learned environment-agnostic representations, the agent is further prevented from the overfitting issue and generalizes better on unseen environments.

To the best of our knowledge, we are the first to introduce natural language grounded multitask and environment-agnostic training regimes and validate their effectiveness on VLN and NDH tasks. Extensive experiments demonstrate that our environment-agnostic multitask navigation model can not only efficiently execute different language guidance in indoor environments but also outperform the single-task baseline models by a large margin on both tasks. Besides, the performance gap between seen and unseen environments is significantly reduced. We also set a new state of the art on NDH with over $120\%$ improvement in terms of goal progress.

## 2 BACKGROUND

**Vision-and-Language Navigation.** Vision-and-Language Navigation (Anderson et al., 2018b; Chen et al., 2019) task requires an embodied agent to navigate in photo-realistic environments to carry out natural language instructions. The agent is spawned at an initial pose $p_0 = (v_0, \phi_0, \theta_0)$, which includes the spatial location, heading and elevation angles. Given a natural language instruction $X = \{x_1, x_2, ..., x_n\}$, the agent is expected to perform a sequence of actions $\{a_1, a_2, ..., a_T\}$ and arrive at the target position $v_{tar}$ specified by the language instruction $X$, which describes step-by-step instructions from the starting position to the target position. In this work, we consider VLN task defined for Room-to-Room (R2R) (Anderson et al., 2018b) dataset which contains instruction-trajectory pairs across 90 different indoor environments (houses).

Previous VLN methods have studied various aspects to improve the navigation performance, such as planning (Wang et al., 2018), data augmentation (Fried et al., 2018; Tan et al., 2019), cross-modal alignment (Wang et al., 2019; Huang et al., 2019b), progress estimation (Ma et al., 2019a), error correction (Ma et al., 2019b; Ke et al., 2019), interactive language assistance (Nguyen et al., 2019; Nguyen & Daumé III, 2019) etc. This work tackles VLN via multitask learning and environment-agnostic learning, which is orthogonal to all these prior arts.

**Navigation from Dialog History.** Different from Visual Dialog (Das et al., 2017) which involves dialog grounded in a single image, the recently introduced Cooperative Vision-and-Dialog Navigation (CVDN) dataset (Thomason et al., 2019) includes interactive language assistance for indoor navigation, which consists of over 2,000 embodied, human-human dialogs situated in photo-realistic home environments. The task of Navigation from Dialog History (NDH) is defined as: given a target object $t_0$ and a dialog history between humans cooperating to perform the task, the embodied agent must infer navigation actions towards the goal room that contains the target object. The dialog history is denoted as $< t_0, Q_1, A_1, Q_2, A_2, ..., Q_i, A_i >$, including the target object $t_0$, the questions $Q$ and answers $A$ till the turn $i$ ($0 \le i \le k$, where $k$ is the total number of Q-A turns from the beginning to the goal room). The agent, located in $p_0$, is trying to move closer to the goal room by inferring from the dialog history that happened before.

**Multitask Learning.** The basis of Multitask (MT) learning is the notion that tasks can serve as mutual sources of inductive bias for each other (Caruana, 1993). When multiple tasks are trained jointly, MT learning causes the learner to prefer the hypothesis that explains all the tasks simultaneously, hence leading to more generalized solutions. MT learning has been successful in natural language processing (Collobert & Weston, 2008), speech recognition (Deng et al., 2013), computer vision (Girshick, 2015), drug discovery (Ramsundar et al., 2015), and Atari games (Teh et al., 2017). The deep reinforcement learning methods that have become very popular for training models on natural language grounded navigation tasks (Wang et al., 2019; Huang et al., 2019a;b; Tan et al., 2019) are known to be data inefficient. In this work, we introduce multitask reinforcement learning for such tasks to improve data efficiency by positive transfer across related tasks.

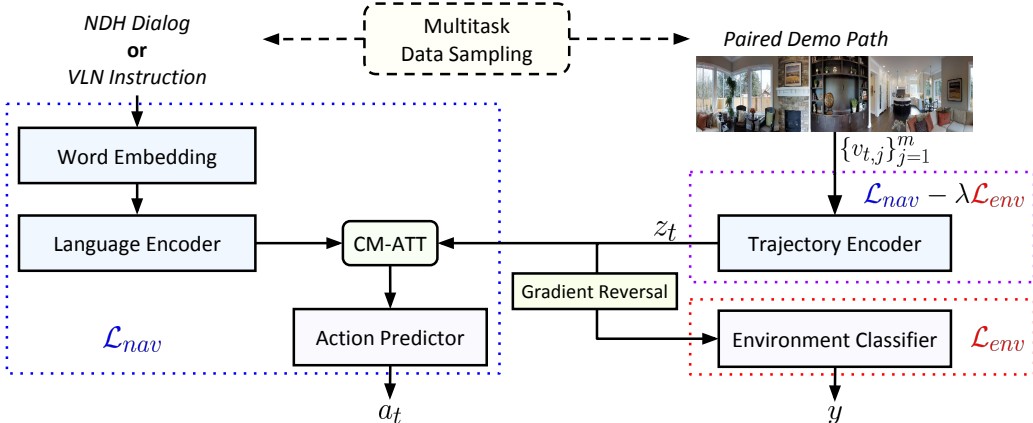

Figure 1: Overview of environment-agnostic multitask learning. See Section 3.1 for more details.

**Environment-agnostic Learning.** A few studies on agnostic learning have been proposed recently. For example, Model-Agnostic Meta-Learning (MAML) (Finn et al., 2017) aims to train a model on a variety of learning tasks and solve a new task using only a few training examples. Liu et al. (2018) proposes a unified feature disentangler that learns domain-invariant representation across multiple domains for image translation. Other domain-agnostic techniques are also proposed for supervised (Li et al., 2018) and unsupervised domain adaption (Romijnders et al., 2019; Peng et al., 2019). In this work, we pair the environment classifier with a gradient reversal layer (Ganin & Lempitsky, 2015) to learn an environment-agnostic representation that can be better generalized on unseen environments in a zero-shot fashion where no adaptation is involved.

**Distributed Actor-Learner Navigation Learning Framework.** To train models for the various language grounded navigation tasks like VLN and NDH, we develop a distributed actor-learner learning infrastructure[1]. The framework design is inspired by IMPALA (Espeholt et al., 2018) and uses its off-policy correction method called V-trace to efficiently scale reinforcement learning methods to thousands of machines. The framework additionally supports a variety of supervision strategies important for navigation tasks such as teacher-forcing (Anderson et al., 2018b), student-forcing (Anderson et al., 2018b) and mixed supervision (Thomason et al., 2019). The framework is built using TensorFlow (Abadi et al., 2016) and supports ML accelerators (GPU, TPU).

## 3 ENVIRONMENT-AGNOSTIC MULTITASK LEARNING

### 3.1 OVERVIEW

Our environment-agnostic multitask navigation model is illustrated in Figure 1. First, we adapt the reinforced cross-modal matching (RCM) model (Wang et al., 2019) and make it seamlessly transfer across tasks by sharing all the learnable parameters for both NDH and VLN, including joint word embedding layer, language encoder, trajectory encoder, cross-modal attention module (CM-ATT), and action predictor. Furthermore, to learn the environment-agnostic representation $z_t$, we equip the navigation model with an environment classifier whose objective is to predict which house the agent is. But note that between trajectory encoder and environment classifier, a gradient reversal layer (Ganin & Lempitsky, 2015) is introduced to reverse the gradients back-propagated to the trajectory encoder, making it learn representations that are environment-agnostic and thus more generalizable in unseen environments. During training, the environment classifier is minimizing the environment classification loss $\mathcal{L}_{env}$, while the trajectory encoder is maximizing $\mathcal{L}_{env}$ and minimizing the navigation loss $\mathcal{L}_{nav}$. The other modules are optimized with the navigation loss $\mathcal{L}_{nav}$ simultaneously. Below we introduce multitask reinforcement learning and environment-agnostic representation learning. A more detailed model architecture is presented in Section 4.

---

[1]The identity is not disclosed to respect the anonymity of the submission.

## 3.2 MULTITASK REINFORCEMENT LEARNING

**Interleaved Multitask Data Sampling.** To avoid overfitting to individual tasks, we adopt an interleaved multitask data sampling strategy to train the model. Particularly, each data sample within a mini-batch can be from either task, so that the VLN instruction-trajectory pairs and NDH dialog-trajectory pairs are interleaved in a mini-batch though they may have different learning objectives.

**Reward Shaping.** Following prior art (Wang et al., 2018; 2019), we first implement a discounted cumulative reward function $R$ for the VLN and NDH tasks:

$$R(s_t, a_t) = \sum_{t'=t}^{T} \gamma^{t'-t} r(s_{t'}, a_{t'}), \text{ where } r(s_{t'}, a_{t'}) = \begin{cases} d(s_{t'}, v_{tar}) - d(s_{t'+1}, v_{tar}) & \text{if } t' < T \\ \mathbb{1}[d(s_T, v_{tar}) \leq d_{th}] & \text{if } t' = T \end{cases} \tag{1}$$

where $\gamma$ is the discounted factor, $d(s_{t'}, v_{tar})$ is the distance between state $s_t$ and the target location $v_{tar}$, and $d_{th}$ is the maximum distance from $v_{tar}$ that the agent is allowed to terminate for success.

Different from VLN, NDH is essentially room navigation instead of point navigation because the agent is expected to reach a room that contains the target object. Suppose the goal room is occupied by a set of nodes $\{v_i\}_1^N$, we replace the distance function $d(s_t, v_{tar})$ in Equation 1 with the minimum distance to the goal room $d_{room}(s_t, \{v_i\}_1^N)$ for NDH:

$$d_{room}(s_t, \{v_i\}_1^N) = \min_{1 \leq i \leq N} d(s_t, v_i) \tag{2}$$

**Navigation Loss.** Since human demonstrations are available for both VLN and NDH tasks, we use behavior cloning to constrain the learning algorithm to model state-action spaces that are most relevant to each task. Following previous works (Wang et al., 2019), we also use reinforcement learning to aid the agent's ability to recover from erroneous actions in unseen environments. During multitask navigation model training, we adopt a mixed training strategy of reinforcement learning and behavior cloning, so the navigation loss function is:

$$\mathcal{L}_{nav} = -\mathbb{E}_{a_t \sim \pi}[R(s_t, a_t) - b] - \mathbb{E}[\log \pi(a_t^* | s_t)] \tag{3}$$

where we use REINFORCE policy gradients (Williams, 1992) and supervised learning gradients to update the policy $\pi$. $b$ is the estimated baseline to reduce the variance and $a_t^*$ is the human demonstrated action.

## 3.3 ENVIRONMENT-AGNOSTIC REPRESENTATION LEARNING

To further improve the generalizability of the navigation policy, we propose to learn a latent environment-agnostic representation that is invariant among seen environments. We would like to get rid of the environment-specific features that are irrelevant to general navigation (e.g. unique house appearances), preventing the model from overfitting to specific seen environments. We can reformulate the navigation policy as

$$\pi(a_t | s_t) = p(a_t | z_t, s_t) p(z_t | s_t) \tag{4}$$

where $z_t$ is a latent representation.

As shown in Figure 1, $p(a_t | z_t, s_t)$ is modeled by the policy module (including CM-ATT and action predictor) and $p(z_t | s_t)$ is modeled by the trajectory encoder. In order to learn the environment-agnostic representation, we employ an environment classifier and a gradient reversal layer (Ganin & Lempitsky, 2015). The environment classifier is parameterized to predict the identity of the house where the agent is, so its loss function $\mathcal{L}_{env}$ is defined as

$$\mathcal{L}_{env} = -\mathbb{E}[\log p(y = y^* | z_t)] \tag{5}$$

where $y^*$ is the ground-truth house label. The gradient reversal layer has no parameters. It acts as an identity transform during forward-propagation, but multiplies the gradient by $-\lambda$ and passes it to the trajectory encoder during back-propagation. Therefore, in addition to minimizing the navigation loss $\mathcal{L}_{nav}$, the trajectory encoder is also maximizing the environment classification loss $\mathcal{L}_{env}$, trying to increase the entropy of the classifier in an adversarial learning manner where the classifier is minimizing the classification loss conditioned on the latent representation $z_t$.

# 4 MODEL ARCHITECTURE

**Language Encoder.** The natural language guidance (instruction or dialog) is tokenized and embedded into n-dimensional space $\boldsymbol{X} = \{\boldsymbol{x}_1, \boldsymbol{x}_2, ..., \boldsymbol{x}_3\}$ where the word vectors $\boldsymbol{x}_i$ are initialized randomly. The vocabulary is restricted to tokens that occur at least five times in the training instructions (The vocabulary used when jointly training VLN and NDH tasks is the union of the two tasks' vocabularies.). All out-of-vocabulary tokens are mapped to a single out-of-vocabulary identifier. The token sequence is encoded using a bi-directional LSTM (Schuster & Paliwal, 1997) to create $\boldsymbol{H}^{\boldsymbol{X}}$ following:

$$\boldsymbol{H}^{\boldsymbol{X}} = [\boldsymbol{h}_1^{\boldsymbol{X}}; \boldsymbol{h}_2^{\boldsymbol{X}}; ...; \boldsymbol{h}_n^{\boldsymbol{X}}], \quad \boldsymbol{h}_t^{\boldsymbol{X}} = \sigma(\overrightarrow{\boldsymbol{h}}_t^{\boldsymbol{X}}, \overleftarrow{\boldsymbol{h}}_t^{\boldsymbol{X}}) \tag{6}$$

$$\overrightarrow{\boldsymbol{h}}_t^{\boldsymbol{X}} = LSTM(\boldsymbol{x}_t, \overrightarrow{\boldsymbol{h}}_{t-1}^{\boldsymbol{X}}), \quad \overleftarrow{\boldsymbol{h}}_t^{\boldsymbol{X}} = LSTM(\boldsymbol{x}_t, \overleftarrow{\boldsymbol{h}}_{t+1}^{\boldsymbol{X}}) \tag{7}$$

where $\overrightarrow{\boldsymbol{h}}_t^{\boldsymbol{X}}$ and $\overleftarrow{\boldsymbol{h}}_t^{\boldsymbol{X}}$ are the hidden states of the forward and backward LSTM layers at time step $t$ respectively, and the $\sigma$ function is used to combine $\overrightarrow{\boldsymbol{h}}_t^{\boldsymbol{X}}$ and $\overleftarrow{\boldsymbol{h}}_t^{\boldsymbol{X}}$ into $\boldsymbol{h}_t^{\boldsymbol{X}}$.

**Trajectory Encoder.** Similar to benchmark models (Fried et al., 2018; Wang et al., 2019; Huang et al., 2019b), at each time step $t$, the agent perceives a 360-degree panoramic view at its current location. The view is discretized into $k$ view angles ($k = 36$ in our implementation, 3 elevations by 12 headings at 30-degree intervals). The image at view angle $i$, heading angle $\phi$ and elevation angle $\theta$ is represented by a concatenation of the pre-trained CNN image features with the 4-dimensional orientation feature [sin $\phi$; cos $\phi$; sin $\theta$; cos $\theta$] to form $\boldsymbol{v}_{t,i}$. The visual input sequence $\boldsymbol{V} = \{\boldsymbol{v}_1, \boldsymbol{v}_2, ..., \boldsymbol{v}_m\}$ is encoded using a LSTM to create $\boldsymbol{H}^{\boldsymbol{V}}$ following:

$$\boldsymbol{H}^{\boldsymbol{V}} = [\boldsymbol{h}_1^{\boldsymbol{V}}; \boldsymbol{h}_2^{\boldsymbol{V}}; ...; \boldsymbol{h}_m^{\boldsymbol{V}}], \quad \text{where } \boldsymbol{h}_t^{\boldsymbol{V}} = LSTM(\boldsymbol{v}_t, \boldsymbol{h}_{t-1}^{\boldsymbol{V}}) \tag{8}$$

$\boldsymbol{v}_t = \text{Attention}(\boldsymbol{h}_{t-1}^{\boldsymbol{V}}, \boldsymbol{v}_{t,1..k})$ is the attention-pooled representation of all view angles using previous agent state $\boldsymbol{h}_{t-1}$ as the query. We use the dot-product attention (Vaswani et al., 2017) hereafter.

**Policy Module.** The policy module comprises of cross-modal attention (CM-ATT) unit as well as an action predictor. The agent learns a policy $\pi_\theta$ over parameters $\theta$ that maps the natural language instruction $\boldsymbol{X}$ and the initial visual scene $\boldsymbol{v}_1$ to a sequence of actions $[a_1, a_2, ..., a_n]$. The action space which is common to VLN and NDH tasks consists of navigable directions from the current location. The available actions at time $t$ are denoted as $\boldsymbol{u}_{t,1..l}$, where $\boldsymbol{u}_{t,j}$ is the representation of the navigable direction $j$ from the current location obtained similarly to $\boldsymbol{v}_{t,i}$. The number of available actions, $l$, varies per location, since graph node connectivity varies. As in Wang et al. (2019), the model predicts the probability $p_d$ of each navigable direction $d$ using a bilinear dot product:

$$p_d = \text{softmax}([\boldsymbol{h}_t^{\boldsymbol{V}}; \boldsymbol{c}_t^{\text{text}}; \boldsymbol{c}_t^{\text{visual}}] \boldsymbol{W}_c (\boldsymbol{u}_{t,d} \boldsymbol{W}_u)^T) \tag{9}$$

where $\boldsymbol{c}_t^{\text{text}} = \text{Attention}(\boldsymbol{h}_t^{\boldsymbol{V}}, \boldsymbol{h}_{1..n}^{\boldsymbol{X}})$ and $\boldsymbol{c}_t^{\text{visual}} = \text{Attention}(\boldsymbol{c}_t^{\text{text}}, \boldsymbol{v}_{t,1..k})$. $\boldsymbol{W}_c$ and $\boldsymbol{W}_u$ are learnable parameters.

**Environment Classifier.** The environment classifier is a two-layer perceptron with a SoftMax layer as the last layer. Given the latent representation $\boldsymbol{z}_t$ (which is $\boldsymbol{h}_t^{\boldsymbol{V}}$ in our setting), the classifier generates a probability distribution over the house labels.

# 5 EXPERIMENTS

## 5.1 EXPERIMENTAL SETUP

**Implementation Details.** In the experiments, we use a 2-layer bi-directional LSTM for the instruction encoder where the size of LSTM cells is 256 units in each direction. The inputs to the encoder are 300-dimensional embeddings initialized randomly. For the visual encoder, we use a 2-layer LSTM with a cell size of 512 units. The encoder inputs are image features derived as mentioned in Section 4. The cross-modal attention layer size is 128 units. The environment classifier has one hidden layer of size 128 units followed by an output layer of size equal to the number of classes. During training, some episodes in the batch are identical to available human demonstrations in the training dataset where the objective is to increase the agent's likelihood of choosing human actions (behavioral cloning (Bain & Sammut, 1999)). The rest of the episodes are constructed by sampling

Table 1: Comparison of agent performance under different training strategies. Note that the single-task RCM model is independently trained and tested on VLN or NDH tasks.

| Model | Val Seen | | | | | | Val Unseen | | | | | |
|---|---|---|---|---|---|---|---|---|---|---|---|---|
| | NDH | VLN | | | | | NDH | VLN | | | | |
| | Progress ↑ | PL | NE ↓ | SR ↑ | SPL ↑ | CLS ↑ | Progress ↑ | PL | NE ↓ | SR ↑ | SPL ↑ | CLS ↑ |
| seq2seq (Thomason et al., 2019) | 5.92 | | | | | | 2.10 | | | | | |
| RCM (Wang et al., 2019)[2] | | 12.08 | **3.25** | **67.60** | - | - | | 15.00 | 6.02 | 40.60 | - | - |
| **Ours** | | | | | | | | | | | | |
| single-task RCM | **6.49** | 10.75 | 5.09 | 52.39 | 48.86 | 63.91 | 2.64 | 10.60 | 6.10 | 42.93 | 38.88 | 54.86 |
| single-task RCM + EnvAg | 6.07 | 11.31 | 4.93 | 52.79 | 48.85 | 63.26 | 3.15 | 11.36 | 5.79 | 44.40 | 40.30 | 55.77 |
| MT-RCM | 5.28 | 10.63 | 5.09 | **56.42** | **49.67** | **68.28** | 4.36 | 10.23 | **5.31** | 46.20 | **44.19** | 54.99 |
| MT-RCM + EnvAg | 5.07 | 11.60 | **4.83** | 53.30 | 49.39 | 64.10 | **4.65** | 12.05 | 5.41 | **47.22** | 41.80 | **56.22** |

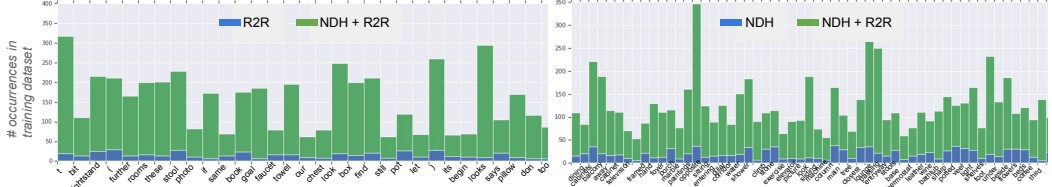

Figure 2: Selected tokens from the vocabulary for VLN (left) and NDH (right) tasks which gained more than 40 additional occurrences in the training dataset due to joint-training.

from agent's own policy. In the experiments, unless otherwise stated, we use entire dialog history from NDH task for model training. *All the reported results in subsequent studies are averages of at least 3 independent runs.*

**Evaluation Metrics.** The agents are evaluated on two datasets, namely *Validation Seen* that contains new paths from the training environments and *Validation Unseen* that contains paths from previously unseen environments. The evaluation metrics for VLN task are as follows: *Path Length (PL)* measures the total length of the predicted path; *Navigation Error (NE)* measures the distance between the last nodes in the predicted and the reference paths; *Success Rate (SR)* measures how often the last node in the predicted path is within some threshold distance of the last node in the reference path; *Success weighted by Path Length (SPL)* (Anderson et al., 2018a) measures Success Rate weighted by the normalized Path Length; and *Coverage weighted by Length Score (CLS)* (Jain et al., 2019) measures predicted path's conformity to the reference path weighted by length score. For NDH task, the agent's progress is defined as reduction (in meters) from the distance to the goal region at agent's first position versus at its last position (Thomason et al., 2019).

## 5.2 Environment-Agnostic Multitask Learning

Table 1 shows the results of training the navigation model using environment-agnostic learning (*EnvAg*) as well as multitask learning (*MT-RCM*). First, both learning methods independently help the agent learn more generalized navigation policy as is evidenced by significant reduction in agent's performance gap between seen and unseen environments. For instance, performance gap for agent's goal progress on NDH task drops from 3.85m to 0.92m using multitask learning and agent's success rate on VLN task between seen and unseen datasets drops from 9.26% to 8.39% using environment-agnostic learning. Second, the two techniques are complementary—the agent's performance when trained with both the techniques simultaneously improves on unseen environments compared to when trained separately. Finally, we note here that *MT-RCM + EnvAg* outperforms the state-of-the-art goal progress of 2.10m (Thomason et al., 2019) on NDH validation unseen dataset by more than 120%. At the same time, it outperforms the equivalent RCM baseline (Wang et al., 2019) of 40.6% success rate by more than 16% (relative measure) on VLN validation unseen dataset.

## 5.3 Multitask Learning

Next, we conduct studies to examine cross-task transfer using multitask learning alone. One of the main advantages of multitask learning is that under-represented tokens in each of the individual tasks

---

[2]We report the performance of the equivalent RCM model without intrinsic reward as the benchmark.

Table 2: Comparison of agent performance when trained separately *vs.* jointly on VLN and NDH.

| Fold | Model | | Inputs for NDH | | | Progress ↑ | PL | NE ↓ | SR ↑ | SPL ↑ | CLS ↑ |
|------|-------|-------|-----|-----|---------------------|------------|----|------|------|-------|-------|
| | | $t_o$ | $A_i$ | $Q_i$ | $A_{1:i-1};Q_{1:i-1}$ | | | | | | |
| Val Seen | NDH-RCM | ✓ | | | | **6.97** | | | | | |
| | | ✓ | ✓ | | | **6.92** | | | | | |
| | | ✓ | ✓ | ✓ | | **6.47** | | | | | |
| | | ✓ | ✓ | ✓ | ✓ | **6.49** | | | | | |
| | VLN-RCM | | | | | | 10.75 | 5.09 | 52.39 | 48.86 | 63.91 |
| | MT-RCM | ✓ | | | | 3.00 | 11.73 | 4.87 | 54.56 | 52.00 | 65.64 |
| | | ✓ | ✓ | | | 5.92 | 11.12 | 4.62 | 54.89 | **52.62** | 66.05 |
| | | ✓ | ✓ | ✓ | | 5.43 | 10.94 | **4.59** | 54.23 | 52.06 | 66.93 |
| | | ✓ | ✓ | ✓ | ✓ | 5.28 | 10.63 | 5.09 | **56.42** | 49.67 | **68.28** |
| Val Unseen | NDH-RCM | ✓ | | | | 1.25 | | | | | |
| | | ✓ | ✓ | | | 2.69 | | | | | |
| | | ✓ | ✓ | ✓ | | 2.69 | | | | | |
| | | ✓ | ✓ | ✓ | ✓ | 2.64 | | | | | |
| | VLN-RCM | | | | | | 10.60 | 6.10 | 42.93 | 38.88 | 54.86 |
| | MT-RCM | ✓ | | | | **1.69** | 13.12 | 5.84 | 42.75 | 38.71 | 53.09 |
| | | ✓ | ✓ | | | **4.01** | 11.06 | 5.88 | 42.98 | 40.62 | 54.30 |
| | | ✓ | ✓ | ✓ | | **3.75** | 11.08 | 5.70 | 44.50 | 39.67 | 54.95 |
| | | ✓ | ✓ | ✓ | ✓ | **4.36** | 10.23 | **5.31** | **46.20** | **44.19** | 54.99 |

Table 3: Comparison of agent performance when language instructions are encoded by separate *vs.* shared encoder for VLN and NDH tasks.

| Language Encoder | Val Seen | | | | | | Val Unseen | | | | | |
|------------------|----------|----|------|------|-------|-------|------------|----|------|------|-------|-------|
| | NDH | VLN | | | | | NDH | VLN | | | | |
| | Progress ↑ | PL | NE ↓ | SR ↑ | SPL ↑ | CLS ↑ | Progress ↑ | PL | NE ↓ | SR ↑ | SPL ↑ | CLS ↑ |
| Shared | **5.28** | 10.63 | 5.09 | **56.42** | **49.67** | **68.28** | **4.36** | 10.23 | **5.31** | **46.20** | **44.19** | **54.99** |
| Separate | 5.17 | 11.26 | **5.02** | 52.38 | 48.80 | 64.19 | 4.07 | 11.72 | 6.04 | 43.64 | 39.49 | 54.57 |

get a significant boost in the number of training samples. Figure 2 illustrates that tokens with less than 40 occurrences end up with sometimes more than 300 occurrences during joint-training.

To examine the impact of dialog history in NDH task, we conduct studies with access to different parts of the dialog—the target object $t_o$, the last oracle answer $A_i$, the prefacing navigator question $Q_i$ and the full dialog history. Table 2 shows the results of jointly training *MT-RCM* model on VLN and NDH tasks. *MT-RCM* model learns a generalized policy that consistently outperforms the competing model with access to similar parts of the dialog on previously unseen environments. As noted before, multitask learning significantly reduces the gap between the agent's performance on previously seen and unseen environments for both tasks. Furthermore, we see a consistent and gradual increase in the success rate of *MT-RCM* on VLN task as it is trained on paths with richer dialog history from the NDH task. This shows that the agent benefits from more complete information about the path implying the importance given by the agent to the language instructions in the task.

We also investigate the impact of parameter sharing of the language encoder for both tasks. As shown in Table 3, the model with shared language encoder for NDH and VLN tasks outperforms the model that has separate language encoders for the two tasks, hence demonstrating the importance of parameter sharing during multitask learning. A more detailed analysis can be found in the Appendix.

## 5.4 ENVIRONMENT-AGNOSTIC LEARNING

From Table 1, it can be seen that both VLN and NDH tasks benefit from environment-agnostic learning independently. To further examine the generalization property due to environment-agnostic objective, we train a model with the opposite objective—learn to correctly predict the navigation environments by removing the gradient reversal layer (*environment-aware learning*). Interesting

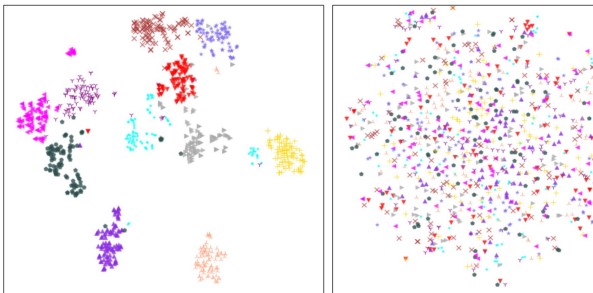

Figure 3: t-SNE visualization of trajectory encoder's output (1000 random paths across 11 different color-coded environments) for models trained with environment-aware objective (left) *versus* environment-agnostic objective (right).

Table 4: Environment-agnostic *versus* environment-aware learning.

(a) Comparison on NDH.

| Model | Val Seen | Val Unseen |
| --- | --- | --- |
| | Progress ↑ | Progress ↑ |
| RCM | 6.49 | 2.64 |
| EnvAware | **8.38** | 1.81 |
| EnvAg | 6.07 | **3.15** |

(b) Comparison on VLN.

| Model | Val Seen | | | | | Val Unseen | | | | |
| --- | --- | --- | --- | --- | --- | --- | --- | --- | --- | --- |
| | PL | NE↓ | SR↑ | SPL↑ | CLS↑ | PL | NE↓ | SR↑ | SPL↑ | CLS↑ |
| RCM | 10.75 | 5.09 | 52.39 | 48.86 | 63.91 | 10.60 | 6.10 | 42.93 | 38.88 | 54.86 |
| EnvAware | 10.30 | **4.36** | **57.59** | **54.05** | **68.49** | 10.13 | 6.30 | 38.83 | 35.65 | 54.79 |
| EnvAg | 11.31 | 4.93 | 52.79 | 48.85 | 63.26 | 11.36 | **5.79** | **44.40** | **40.30** | **55.77** |

results are observed in Table 4 that environment-aware learning leads to overfitting on the training dataset (performance on environments seen during training consistently increases for both tasks), while environment-agnostic learning leads to more generalizable policy which performs better on previously unseen environments. Figure 3 further shows that due to environment-aware objective, the model learns to represent visual inputs from the same environment closer to each other while the representations of different environments are farther from each other resulting in a clustering learning effect. On the other hand, the environment-agnostic objective leads to more general representation across different environments which results in better performance on unseen environments.

## 5.5 REWARD SHAPING FOR NDH TASK

As discussed in Section 3.2, we conducted studies to shape the reward for NDH task. The results in Table 5 indicate that incentivizing the agent to get closer to the goal room is better than to the exact goal location, because it is aligned with the objective of NDH task, which is to reach the room containing the goal object. Detailed ablation is presented in Appendix showing that the same holds true consistently as the agent is provided access to different parts of the dialog history.

Table 5: Average agent progress towards goal room when trained using different rewards.

| Model | Goal Progress (m) | |
| --- | --- | --- |
| | Val Seen | Val Unseen |
| Shortest-Path Agent | 9.52 | 9.58 |
| Random Agent | 0.42 | 1.09 |
| seq2seq (Thomason et al., 2019) | 5.92 | 2.10 |
| **Ours** | | |
| NDH-RCM (dis to goal location) | 5.02 | 2.58 |
| NDH-RCM (dis to goal room) | **6.49** | **2.64** |

## 6 CONCLUSION

In this work, we show that the model trained using environment-agnostic multitask learning approach learns a generalized policy for the two natural language grounded navigation tasks. It closes down the gap between seen and unseen environments, learns more generalized environment representations and effectively transfers knowledge across tasks outperforming baselines on both the tasks simultaneously by a significant margin. At the same time, the two approaches independently benefit the agent learning and are complementary to each other. There are possible future extensions to our work—the *MT-RCM* can further be adapted to other language-grounded navigation datasets, such as those using Street View (e.g., Touchdown (Chen et al., 2019), TalkTheWalk (de Vries et al., 2018)); and complementary techniques like environmental dropout (Tan et al., 2019) can be combined with environment-agnostic learning to learn more general representations.

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

## A APPENDIX

### A.1 REWARD SHAPING FOR NDH TASK

Table 6 presents a more detailed ablation of Table 5 using different parts of dialog history. The results prove that agents rewarded for getting closer to the goal room consistently outperform agents rewarded for getting closer to the exact goal location.

Table 6: Average agent progress towards goal room when trained using different rewards and mixed supervision strategy.

| | Model | $t_0$ | $A_i$ | $Q_i$ | $A_{1:i-1}; Q_{1:i-1}$ | Val Seen | Val Unseen |
|---|---|---|---|---|---|---|---|
| | | | | **Inputs** | | **Goal Progress (m)** | |
| Baselines | Shortest-Path Agent | | | | | 9.52 | 9.58 |
| | Random Agent | | | | | 0.42 | 1.09 |
| | Seq-2-Seq (Thomason et al., 2019) | ✓ | | | | 5.71 | **1.29** |
| | | ✓ | ✓ | | | 6.04 | 2.05 |
| | | ✓ | ✓ | ✓ | | 6.16 | 1.83 |
| | | ✓ | ✓ | ✓ | ✓ | 5.92 | 2.10 |
| Ours | NDH-RCM (distance to goal location) | ✓ | | | | 4.18 | 0.42 |
| | | ✓ | ✓ | | | 4.96 | 2.34 |
| | | ✓ | ✓ | ✓ | | 4.60 | 2.25 |
| | | ✓ | ✓ | ✓ | ✓ | 5.02 | 2.58 |
| | NDH-RCM (distance to goal room) | ✓ | | | | **6.97** | 1.25 |
| | | ✓ | ✓ | | | **6.92** | **2.69** |
| | | ✓ | ✓ | ✓ | | **6.47** | **2.69** |
| | | ✓ | ✓ | ✓ | ✓ | **6.49** | **2.64** |

### A.2 DETAILED ABLATION ON PARAMETER SHARING OF LANGUAGE ENCODER

Table 7 presents a more detailed analysis from Table 3 with access to different parts of dialog history. The models with shared language encoder consistently outperform those with separate encoders.

Table 7: Comparison of agent performance when language instructions are encoded by separate *vs.* shared encoder for VLN and NDH tasks. All the reported results are averages of 3 independent runs.

| Fold | Language Encoder | $t_0$ | $A_i$ | $Q_i$ | $A_{1:i-1}; Q_{1:i-1}$ | Goal Progress ↑ | PL | NE ↓ | SR ↑ | SPL ↑ | CLS ↑ |
|---|---|---|---|---|---|---|---|---|---|---|---|
| | | | | **Inputs for NDH** | | **NDH Evaluation** | | | **VLN Evaluation** | | |
| Val Seen | Shared | ✓ | | | | **3.00** | 11.73 | 4.87 | 54.56 | 52.00 | 65.64 |
| | | ✓ | ✓ | | | **5.92** | 11.12 | **4.62** | 54.89 | **52.62** | 66.05 |
| | | ✓ | ✓ | ✓ | | **5.43** | 10.94 | 4.59 | 54.23 | 52.06 | 66.93 |
| | | ✓ | ✓ | ✓ | ✓ | **5.28** | 10.63 | 5.09 | **56.42** | 49.67 | **68.28** |
| | Separate | ✓ | | | | 2.85 | 11.43 | 4.81 | 54.66 | 51.11 | 65.37 |
| | | ✓ | ✓ | | | 4.90 | 11.92 | 4.92 | 53.64 | 49.79 | 61.49 |
| | | ✓ | ✓ | ✓ | | 5.07 | 11.34 | 4.76 | 55.34 | 51.59 | 65.52 |
| | | ✓ | ✓ | ✓ | ✓ | 5.17 | 11.26 | 5.02 | 52.38 | 48.80 | 64.19 |
| Val Unseen | Shared | ✓ | | | | 1.69 | 13.12 | 5.84 | 42.75 | 38.71 | 53.09 |
| | | ✓ | ✓ | | | **4.01** | 11.06 | 5.88 | 42.98 | 40.62 | 54.30 |
| | | ✓ | ✓ | ✓ | | **3.75** | 11.08 | 5.70 | 44.50 | 39.67 | 54.95 |
| | | ✓ | ✓ | ✓ | ✓ | **4.36** | 10.23 | **5.31** | **46.20** | **44.19** | 54.99 |
| | Separate | ✓ | | | | **1.79** | 11.85 | 6.01 | 42.43 | 38.19 | 54.01 |
| | | ✓ | ✓ | | | 3.66 | 12.59 | 5.97 | 43.45 | 38.62 | 53.49 |
| | | ✓ | ✓ | ✓ | | 3.51 | 12.23 | 5.89 | 44.40 | 39.54 | 54.55 |
| | | ✓ | ✓ | ✓ | ✓ | 4.07 | 11.72 | 6.04 | 43.64 | 39.49 | 54.57 |

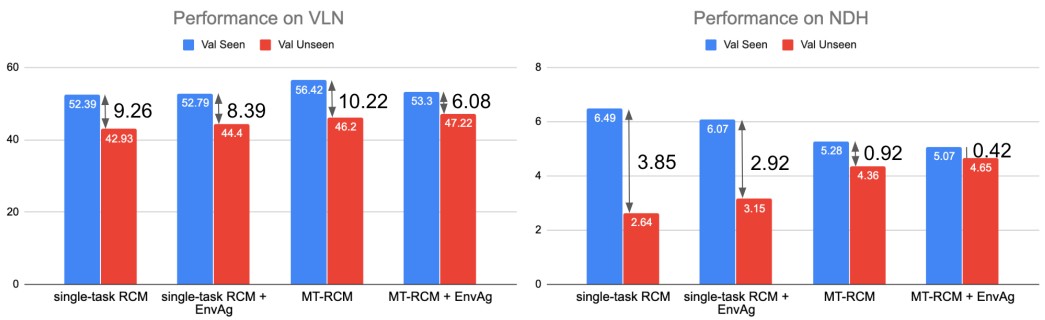

Figure 4: Visualizing performance gap between seen and unseen environments for VLN and NDH tasks. For VLN, the plotted metric is agent's success rate while for NDH, the metric is agent's progress.

### A.3 PERFORMANCE GAP BETWEEN SEEN AND UNSEEN ENVIRONMENTS

As mentioned in Section 5.2, both multitask learning as well as environment-agnostic learning methods reduce the agent's performance gap between seen and unseen environments which is demonstrated in Figure 4.

