# OpenReview forum: "Generalized Natural Language Grounded Navigation via Environment-agnostic Multitask Learning"
_ICLR.cc/2020/Conference — Reject_

### Official Review · AnonReviewer3 · 2019-10-23
**Official Blind Review #3**

**Rating:** 6

**Review:**

This paper addresses some challenges of following natural language instructions for navigating in visual environments. The main challenge in such tasks is the scarcity of available training data, which results in generalization problems where the agent has difficulty navigating in unseen environments. Therefore, the authors propose two key ideas to tackle this issue and incorporate them in the reinforced cross-modal matching (RCM) model (Wang et al, 2019). First, they use a generalized multitask learning model to transfer knowledge across two different tasks: Vision-Language Navigation (VLN) and Navigation from Dialog History (NDH). This results in learning features that explain both tasks simultaneously and hence generalize better. Moreover, by training on both tasks the effective size of training data is increased significantly. Second, they propose an environment-agnostic learning technique in order to learn invariant representations that are still efficient for navigation. This prevents overfitting to specific visual features of the environment and therefore helps improving generalization. The contribution of this paper is combining and incorporating these two key ideas in the RCM framework and verifying it on VLN and NDH tasks. This approach is novel compared to prior results in tackling the VLN task. Their experimental results show that the two proposed techniques improve generalization in a complementary fashion, measured by decreased performance gap between seen and unseen environments.  They demonstrate that their technique outperforms state-of-the-art methods on unseen data on some evaluation metrics.

Even though the two key ideas proposed in the paper has been explored in the literature in other contexts, this paper contributes to natural language guided navigation by incorporating these ideas in a unified framework and demonstrates promising results on two datasets. Therefore, I would recommend accepting this paper if some issues in delivery and clarity are addressed.

In particular, Section 3, where the authors introduce the novelty of the paper in more detail, could be better explained and a cleaner line should be drawn between prior results from other papers and novel results proposed by this paper. In general, the new ideas would require more emphasis, since they are somewhat lost between adaptations from prior work. Most importantly, Eq. (3) is stated without sufficient motivation and would require a more detailed explanation.

In addition to these points, I would like to disclose some recommendations that might improve the paper but are not strictly part of my decision. In some cases the notation is not clear and some variables are not defined or explained. For instance, after Eq. (8) Attention(.) is used without citation or definition, in Eq. (9) Wc and Wu are not defined and some of the notation in Eq. (6)-(7) are not defined. Moreover, reading the decrease in performance gap from the presented table format is inconvenient and a better visual representation might help demonstrating the improvement in generalization better.  Lastly, I noticed that the navigation error for shared decoder is slightly higher than for separate encoders in Table 3, even though it outperforms the separate encoders in every other measures. Is there a particular explanation for this?

**Experience Assessment:**

I do not know much about this area.

**Review Assessment: Checking Correctness Of Derivations And Theory:**

N/A

**Review Assessment: Checking Correctness Of Experiments:**

I assessed the sensibility of the experiments.

**Review Assessment: Thoroughness In Paper Reading:**

I read the paper at least twice and used my best judgement in assessing the paper.

---

> ### Author Response · Authors · 2019-11-15
> **Thanks for your valuable feedback and recommendation of this work**
>
> Thank you for the recommendation and valuable suggestions to improve the clarity of this paper. Below we provide the responses to your suggestions. Note that we have uploaded a paper revision according to the suggestions.
>
> 1. Re: Emphasis on new ideas and results
> We mainly discuss the difference from prior art in Section 2 “background”. As R3 suggested, we explain the motivation of Equation 3 in the revision and draw more clear separation of results of previous work and our new results in Table 1 and Table 5. We will keep working on the clarity till the camera-ready version if it is accepted.
>
> 2. Re: Notations of Equations
> We have improved the notations and citations, and made them clearer in the revision.
>
> 3. Re: Better visualization of performance gaps
> We added a figure (Fig 4) in the Appendix to better visualize the performance gaps between seen and unseen environments in the revision.
>
> 4. Re: Separate encoders vs. shared encoder
> From Table 3, we can observe that the shared encoder outperforms the separate encoders almost on all metrics on both seen and unseen environments. The navigation errors on seen environments are actually equivalent (5.09 vs. 5.02, the difference is 0.07). Besides, the navigation error is the average distance between the last node in the predicted path and the target destination, which is often but not necessarily aligned to the primary measures like Success Rate, because 5 meters and 10 meters to the destination are both considered as failure cases.

---

### Official Review · AnonReviewer2 · 2019-10-23
**Official Blind Review #2**

**Rating:** 3

**Review:**

This paper aims to apply the model of Wang 2019 to the new NDH task of Thomason '19.  Both of these datasets are built on the same room-to-room environment and both are for natural language instruction following.  Thomason's work extends the R2R paradigm to include a dialogue history which is collapsed into a single instruction.  The contribution of this paper is to build a single model which alternatingly samples trajectories from each of the two datasets to train a more general actor and the authors also believe that the presence of an environment classifier assists in generalization.

The claims of the paper focus on being "environment-agnostic" and notions of generality.  As hinted by the authors in their future work, to properly show this would probably require two different environments or tasks (e.g. Touchdown), not training on two tasks that use the same environments and pre-computed visual features.  Am I incorrect that the only difference between the NDH and VLN formulation is the structure of the sentences?

Figure 3 is the most compelling component of the paper.  However, I am still not convinced it will generalize and all other components of the paper are largely minor tweaks to existing work.

Figure 2 mostly leads me to believe that we have a simple data-augmentation situation which makes the bump in performance somewhat predictable.  Minor: Is there any reason why in Table 1 we can't simply run the VLN models on NDH and NDH on VLN?

I commend the authors for putting this all together in the two months between the release of CVDN and the ICLR deadline.
I think training a joint model on these two datasets is a completely natural experiment that many of us wanted to see, and so I appreciate the effort of the authors and the benefit to the community of having these numbers, but I'm not convinced there is that much meat otherwise in this paper.

**Experience Assessment:**

I have published in this field for several years.

**Review Assessment: Checking Correctness Of Derivations And Theory:**

N/A

**Review Assessment: Checking Correctness Of Experiments:**

I carefully checked the experiments.

**Review Assessment: Thoroughness In Paper Reading:**

I read the paper at least twice and used my best judgement in assessing the paper.

---

> ### Author Response · Authors · 2019-11-15
> **Thanks for your review and we hope to clarify some confusion (part 1/2)**
>
> Thank you for the review and acknowledging the usefulness of this work. We would like to use this opportunity to resolve some potential confusion and misunderstanding.
>
> 1. Contributions
> First, we would like to reiterate the main contributions:
> (1) We introduce the first generalized multitask learning framework for natural language grounded navigation tasks such as VLN and NDH, which adopts an interleaved multitask sampling strategy and allows different learning objectives for different tasks. Note that we simultaneously sample data for different tasks within the same mini-batch (not alternately batch by batch) to avoid overfitting to individual tasks.
> (2) To further improve generalizability, we propose an environment-agnostic training method, which is unified with the multitask learning framework, to learn environment-invariant representations and thus reduce the gap between seen and unseen environments.
> (3) We have done thorough and extensive experiments on VLN and NDH tasks and prove that the proposed methods are very effective, which improve the baselines by a large margin and establish new SOTA on NDH (with ~120% improvement in terms of goal progress).
>
> 2. Re: “environments” and “environment-agnostic”
> We would also like to clarify the notion of “environments”. We use the term “environments” to refer to different houses in the datasets, which are split into seen and unseen sets. The navigation models are trained on seen environments but tested on previously unseen environments. The objective is to improve the performance on unseen environments and reduce the performance gap between them.
> Therefore, “environment-agnostic” learning is to learn invariant indoor representations among different houses within tasks such as VLN and NDH (not among different tasks). In contrast, the idea of multitask learning is to utilize knowledge across tasks.
>
> 3. Re: differences between VLN and NDH
> Even though both VLN and NDH tasks use the same Matterport3D indoor environments [1], there are significant differences in the motivations as well as the overall objectives of the two tasks --
> (1) Collecting data for VLN task involved single human player describing a fixed path through the environment resulting in instructions that can be followed step-by-step to reach the destination. On the other hand, collecting data for NDH involved two human players (oracle and navigator) co-operating to find a specified object (e.g., “trashcan”) in the environment. The navigator can elicit assistance from the oracle who can view the navigator’s path so far and the future 5 steps towards the goal. As a result, NDH involves a series of question/answer interactions (dialog) between two players that may not necessarily be step-by-step. NDH is actually a more practical setting than VLN. In real-world applications such as disaster relief, it is impossible to just give a one-time instruction, and send the robot to finish the job, so how to navigate from the interactive dialog history is crucially important.
> (2) The two tasks have different data characteristics, e.g., while the VLN task has an average instruction length of 29 words, the average dialog length in NDH is 81 words. Similarly, the average path length in NDH is much larger (roughly 3x) than that in VLN.
> (3) The two tasks differ in input representations as well as overall objectives. While VLN instructions can be tokenized and represented as word embeddings, NDH instructions require care in adding markers to indicate start/end of a navigator’s question and oracle’s answer. Furthermore, while the objective in VLN is to find the exact goal node in the environment (i.e., point navigation), the objective in NDH is the find the goal room that contains the specified object (i.e., room navigation).
>
> Due to the above differences, multitask learning involving the two tasks aids learning better representations that improve performance on both tasks simultaneously. The difference in data characteristics leads to effective inductive transfer between the two tasks (e.g., MT-RCM model can benefit from following shorter paths in VLN to break down longer paths into smaller achievable chunks in NDH). Extending our work to include different environments (e.g., Touchdown) will indeed be a worthy future extension of our work but that should not discount the contributions of our present work.
>
> [1] “Matterport3D: Learning from RGB-D Data in Indoor Environments”, Chang et al. 2017

---

> > ### Author Response · Authors · 2019-11-15
> > **Response to Reviewer #2 (Cont. part 2/2)**
> >
> > 4. Re: data augmentation causing bumps in performance
> > Generally speaking, implicit data augmentation is one aspect that helps multitask learning (see https://ruder.io/multi-task/index.html#whydoesmtlwork) but it is not the sole contributing factor to make it work. Our results indicate that MT-RCM model’s performance on both tasks improve simultaneously with NDH’s progress improving by 65% over the single-task baseline (from 2.64 to 4.36). At the same time, environment-agnostic learning also leads to significant gains on top of single-task baselines as well as complements multitask learning (MT-RCM + EnvAg). The results indicate that our proposed model leads to better generalization by narrowing the gap between seen and unseen environments due to better representations learned via inductive transfer, regularizing environment representations, and attention focusing.
> >
> > 5. Re: single-task RCM model
> > Sorry for the confusion about the results of the single-task RCM model in Table 1. VLN-RCM and NDH-RCM are the same RCM model, but trained on the two different datasets with different reward functions (as introduced in Section 3.2). We have revised Table 1 in the revision to avoid such confusion.
> >
> > 6. Re: Time
> > Thanks for the compliment! But we would like to point out that we have been working on language grounded navigation (e.g. VLN) even before the release of the CVDN dataset, so it is actually more than two and a half months as R2 stated. More importantly, we do not think the time is relevant to the paper quality here. As Reviewer #1 said, “experiments are thorough and have all the ablations one would ask for”. We would also like to mention that all the numbers reported in the paper are averages of at least three different runs (Section 5.1).

---

### Official Review · AnonReviewer1 · 2019-10-23
**Official Blind Review #1**

**Rating:** 6

**Review:**

Summary:

There have been two recent related tasks proposed in vision-langauge settings: vision-langauge navigation (VLN) where natural language turn-by-turn instructions must be decoded by an agent in an indoor environment to reach the goal location and Navigation from Dialog History (NDH) where dialog between two humans trying to reach a goal is input to an agent to try to reach a goal location. This paper uses these two tasks' data in a multi-task manner to try to generalize better between indoor environments especially unseen environments which are not in the training set of the agent.

Another proposed innovation is to use an auxiliary task of environment classification but via a gradient reversal layer such that the learnt latent representation input to the classifier should not overfit to foibles of the environment but should (hopefully) learn a representation that captures the intrinsics of the environment necessary for generalization.

Comments:

- The paper is well-written and easy to understand! Experiments are thorough and has all the ablations one would ask for. Thanks!

- Overall I like the paper but have a number of comments:

1. Why not try more sophisticated methods of multi-task learning like 'MetaLearning' by Finn et al 2017 (MAML). It is common knowledge that straight up averaging across tasks is not as effective as doing the bilevel optimization in MAML.

2. Why do RL at all? Already the authors are doing BC (naive form of imitation learning) but they could just do robust imitation learning like DAgger, AggreVateD, etc. The setting is already such that one has a natural oracle (which the authors are already using via BC in the objective) which is the shortest path planner during training time. Then combined with MAML one can do Meta-IL as in 'One-Shot Imitation Learning via Meta-Learning' Finn et al 2017. Note that imitation learning is exponenially more sample efficient than RL and removes all the reward-shaping complications.

3. In Section 2 for error correction "Vision-based Navigation with Language-based Assistance via Imitation Learning with Indirect Intervention" by Nguyen et al. CVPR 2019 is directly relevant.

4. Also for generalizaton performance these papers are directly relevant:

"Building Generalizable Agents with a Realistic and Rich 3D Environment" Wu et al ICLR 2018

"Learning and Planning with a Semantic Model" Wu et al






**Experience Assessment:**

I have published one or two papers in this area.

**Review Assessment: Checking Correctness Of Derivations And Theory:**

N/A

**Review Assessment: Checking Correctness Of Experiments:**

I carefully checked the experiments.

**Review Assessment: Thoroughness In Paper Reading:**

I read the paper thoroughly.

---

> ### Author Response · Authors · 2019-11-15
> **Thanks for your constructive review and support of this work**
>
> We appreciate your constructive feedback that brings in deeper thinking and helps improve the comprehension of this paper. The responses to your comments are shown below.
>
> 1. Re: why not try more sophisticated methods of multi-task learning like MAML [1].
>
> One of the focuses of this paper is to validate multitask learning for VLN and NDH, so we believe, focusing on simplicity and effectiveness is an elegant solution to initiate the study of language grounded navigation tasks along this direction.
> MAML [1] has been proved very effective in learning generalizable parameters from multiple tasks that can quickly adapt to a new task (few-shot learning). In this paper, we are solving the standard setup of VLN and NDH, where the model trained on seen environments is supposed to directly applied to unseen environments (similar to zero-shot learning). We agree that adapting MAML on these practical tasks would also be very useful, but we also believe it would not devalue our contributions.
>
> [1] “Model-Agnostic Meta-Learning for Fast Adaptation of Deep Networks”, Finn et al. 2017
>
> 2. Re: why do RL instead of more complicated imitation learning.
> Anderson et al [2] introduce an online version of DAgger, student forcing, which imitates the shortest-path actions but takes actions sampled from its own policy. It has been proven to be more effective than teacher forcing (or behavior cloning), but less effective than RL methods [3, 4]. Note that imitation learning methods tend to utilize underlying navigation graphs of seen environments more, resulting in higher performance on seen but lower performance on unseen environments. In contrast, RL methods seem to generalize better on unseen environments. But if we modify the evaluation setup and allow pre-exploration of the unseen environments before testing, then one-shot imitation learning method [5] would fit and can possibly achieve promising performance.
>
> [2] “Vision-and-Language Navigation: Interpreting visually-grounded navigation instructions in real environments”, Anderson et al. 2018
> [3] “Reinforced cross-modal matching and self-supervised imitation learning for vision-language navigation” Wang et al 2019
> [4] “Learning to Navigate Unseen Environments: Back Translation with Environmental Dropout”, Tan et al. 2019
> [5] “One-Shot Imitation Learning via Meta-Learning”, Finn et al. 2017
>
> 3. Re: more related work.
> Thanks for pointing out [6,7,8] in terms of error correction and generalization, we have added them and other related work like [9] in the revision.
>
> [6] “Vision-based Navigation with Language-based Assistance via Imitation Learning with Indirect Intervention”, Nguyen et al. 2019
> [7] “Building Generalizable Agents with a Realistic and Rich 3D Environment”, Wu et al. 2018
> [8] "Learning and Planning with a Semantic Model" Wu et al. 2018
> [9] “help, anna! vision-based navigation with natural multimodal assistance via retrospective curiosity-encouraging imitation learning”, Nguyen et al. 2019

---

> > ### Comment · AnonReviewer1 · 2019-11-15
> > **RL vs. IL**
> >
> > Would the authors give some more bits on the statements "Note that imitation learning methods tend to utilize underlying navigation graphs of seen environments more, resulting in higher performance on seen but lower performance on unseen environments. In contrast, RL methods seem to generalize better on unseen environments." Are there papers/experiments that show this? Would be quite interesting phenomenon.
> >
> > "But if we modify the evaluation setup and allow pre-exploration of the unseen environments before testing, then one-shot imitation learning method [5] would fit and can possibly achieve promising performance.": I found this statement a bit baffling. Why would one have to allow pre-exploration of unseen environments to make [5] work?

---

> > > ### Author Response · Authors · 2019-11-15
> > > **RL + BC vs. IL**
> > >
> > > 1.
> > > To avoid confusion, we would like to note that the RL methods for VLN we mentioned above [2,3] are MIXER for RCM[2] and RL + BC for EnvDrop[3] (as used in [4]).   In contrast, Speaker-Follower [1] solely uses student forcing. Both RCM[2] and EnvDrop[3] outperform Speaker-Follower [1] on unseen environments.
> > >
> > > If we take a closer look at Table 2 in [3] (IL + RL, more specifically BC + RL) and combine it with Line 6 of Table 1 in [1], we could observe that BC + RL achieves worse performance on seen environments (55.3 vs. 66.4 in terms of success rate) but better performance on unseen ones (46.5 vs. 35.5) than Student Forcing [1].
> > > Technically speaking, RL only would work poorly in VLN due to sample inefficiency.
> > >
> > > We had the same observations in our experiments, and we adopt RL + BC as our navigation objective (Equation 3).
> > >
> > > 2.
> > > Under the standard setup of VLN and NDH, the model is trained on seen environments and directly tested on unseen environments without any adaptation. The model is not allowed to use the meta-information of the unseen environments and no demonstrations are permitted. This zero-shot-fashion setting is different from some popular synthetic environments where adaptation is allowed or a few demonstrations are provided.
> > > Both settings make sense in the real world. We consider demonstrations of the unseen environments as a variant of pre-exploration in them (but it is open for discussion), which is also explored in recent work (e.g. SIL [2]).
> > >
> > > [1] "Speaker-Follower Models for Vision-and-Language Navigation", Fried et al 2018
> > > [2] “Reinforced cross-modal matching and self-supervised imitation learning for vision-language navigation” Wang et al 2019
> > > [3] “Learning to Navigate Unseen Environments: Back Translation with Environmental Dropout”, Tan et al. 2019
> > > [4] "A deep reinforced model for abstractive summarization" Paulus et al. 2018
> > > [5] “One-Shot Imitation Learning via Meta-Learning”, Finn et al. 2017

---

### Decision · Program_Chairs · 2019-12-19

**Decision:**

Reject

**Comment:**

The paper proposes a multitask navigation model that can be trained on both vision-language navigation (VLN) and navigation from dialog history (NDH) tasks. The authors provide experiments that demonstrate that their model can outperformance single-task baseline models.

The paper received borderline scores with two weak accept and one weak reject.  Overall, the reviewers found the paper to be well-written and easy to understand, with thorough experiments.

The reviewers had minor concerns about the following:
1. The generalizability of the work.  No results are reported on the test set, only on val.
2. The gains for val unseen are pretty small and there are other models (e.g. Ke et al, Tan et al) that have better results.  Would the proposed environment-agnostic multitask learning be able to improve those models as well?  Or is the gains limited to having a weak baseline?
3. It's unclear if the gains are due to the multitasking or just having more data available to train on.
4. There are some minor issues with the misspellings/typos.  Some examples are given:
Page 1: "Manolis Savva* et al" --> "Savva et al"
Page 5: "x_1, x2, ..., x_3" --> Should the x_3 be something like x_k where k is the length of the utterance?

The AC agrees with the reviewers that the paper is interesting and is mostly solid work.  The AC also feels that there are some valid concerns about the generalizability of the work and that the paper would benefit from a more careful consideration of the issues raised by the reviewers.  The authors are encouraged to refine the work and resubmit.